# Re-Irradiation for Recurrent Cervical Cancer: A State-of-the-Art Review

Zongyan Shen †, Ang Qu †, Ping Jiang, Yuliang Jiang, Haitao Sun  and Junjie Wang *

Department of Radiation Oncology, Peking University Third Hospital, Beijing 100191, China; szongyan11@outlook.com (Z.S.); qa1980@sina.com (A.Q.); jiangping@bjmu.edu.cn (P.J.); yuliang_jiang@sina.com (Y.J.); haitaos@163.com (H.S.)
* Correspondence: junjiewang_edu@sina.cn; Tel.: +86-189-1059-2802
† These authors contributed equally to this work.

**Abstract:** The recurrence rate of cervical cancer after primary treatment can reach 60%, and a poor prognosis is reported in most cases. Treatment options for the recurrence of cervical cancer mainly depend on the prior treatment regimen and the location of recurrent lesions. Re-irradiation is still considered as a clinical challenge, owing to a high incidence of toxicity, especially in in-field recurrence within a short period of time. Recent advances in radiotherapy have preliminarily revealed encouraging outcomes of re-irradiation. Several centers have concentrasted on stereotactic body radiation therapy (SBRT) for the treatment of well-selected cases. Meanwhile, as the image-guiding techniques become more precise, a better dose profile can also be achieved in brachytherapy, including high-dose-rate interstitial brachytherapy (HDR-ISBT) and permanent radioactive seed implantation (PRSI). These treatment modalities have shown promising efficacy with a tolerable toxicity, providing further treatment options for recurrent cervical cancer. However, it is highly unlikely to draw a definite conclusion from all of those studies due to the large heterogeneity among them and the lack of large-scale prospective studies. This study mainly reviews and summarizes the progress of re-irradiation for recurrent cervical cancer in recent years, in order to provide potential treatment regimens for the management of re-irradiation.

**Keywords:** re-irradiation; recurrent cervical cancer; stereotactic body radiation therapy; high-dose-rate interstitial brachytherapy; permanent radioactive seed implantation



## 1. Introduction

The treatment of primary cervical cancer mainly includes surgery, radiotherapy (RT), and concurrent chemoradiotherapy, and the recurrence rate was reported to be 25–61% after the primary treatment [1]. The recurrence pattern of cervical cancer is divided into pelvic recurrence and extra-pelvic recurrence, of which pelvic recurrence includes a central and peripheral type. Studies have shown that peripheral recurrence has a worse prognosis than central recurrence [2,3]. The central type refers to the recurrent lesion located at the centre or midline of the pelvis, which originates from the retained cervix and vagina following primary radiation, or from the vaginal cuff and central scar after surgery. It could invade anteriorly, posteriorly (bladder, rectum), or laterally (parametria), while it does not reach the pelvic wall. In contrast, peripheral recurrence refers to invasion of the pelvic wall or adhesion to the pelvic wall revealed by clinical examination or imaging [4–6]. Recurrent cervical cancer (rCC) that is confined to the cervix or upper vagina can be treated properly. However, the treatment of recurrence in the rest of the parts of the pelvis remains challenging [1].

The recurrence rate of cervical cancer at the irradiated field after radical or adjuvant radiotherapy is 20–40% [7]. The only possible curative option is pelvic exenteration, and surgery may be preferred for patients with no history of radiotherapy or out-field recurrence or in-field recurrence of central lesions, if indicated. However, this treatment technique

is mainly limited by the severe postoperative complications. Apart from surgery, radical radiotherapy with or without chemotherapy can be used for the retreatment of gynecologic tumors. Currently, intraoperative radiotherapy (IORT) is seen as a first-line choice of RT for previously irradiated rCC which can be performed in either pelvic exenteration or local resection. Individual external beam radiation therapy or brachytherapy are also considered as a choice for re-irradiation. However, it is conventionally believed that re-irradiation is typically associated with a high risk of complications, therefore it should be conducted cautiously [8]. Considering a semi-recovery of normal tissues after the first course of radiotherapy, an optimal dose distribution with a short course of administration is required to protect surrounding normal tissues.

With the development of technologies, such as stereotactic body radiation therapy (SBRT) and image-guided radiation therapy (IGRT), secondary radiotherapy has markedly attracted clinicians' attention in recent years. Re-irradiation depends on a patient's history of radiotherapy (including site, technique, dose, and fraction schedule), survival status, and disease-free survival [9]. However, owing to the lack of treatment experience, oncologists are still concerned about surrounding normal tissues and toxicities. Therefore, specialists need to weigh the pros and cons in order to formulate individualized protocols. This overview aims to review the clinical outcomes of re-irradiation by outlining the recent literature and to summarize how modern radiotherapy techniques can positively participate in the retreatment of pelvic recurrences of cervical cancer.

## 2. Brachytherapy

### 2.1. High-Dose-Rate ISBT (HDR-ISBT)

HDR-ISBT is a type of BT, where needles or catheters are inserted directly into the tumor or into the volume of interest. A high dose is delivered to the target area, while the dose to the surrounding normal tissues is reduced, leading to a more flexible and highly conformal dose distribution. Another advantage is that the treatment course is shorter, with a delivery of higher doses within 5–6 days, and it is more effective in tumors with a short tumor doubling time. Compared to intracavitary irradiation, HDR-ISBT is more advantageous for the treatment of relatively large, deeply involved cervical lesions, tumors extending into the parametria or pelvic sidewall, tumors extending into the lower vaginal canal, and suboptimal anatomy such as narrow vagina apex, etc. [10].

Early studies reported that HDR-BT was as effective as or slightly more effective than surgery [11]. Traditionally, the dose of BT was set based on the "Manchester system", which did not take into account a patient's anatomy. In the last decade, BT has been spurred by significant technological advances, and the GYN GECESTRO working group from Europe has published a series of recommendations, defining procedures for image-based BT planning [12–16]. Imaging techniques, such as computed tomography (CT) and magnetic resonance imaging (MRI), have improved visualization of the target area and the organs at risk (OARs), hence, three-dimensional (3D) image-guided BT (3D-IGBT) has emerged. The tumor and OARs can be accurately depicted, thereby improving the dose profile. The outcomes of cervical cancer in terms of both LC rate and toxicity are thus improved. The studies including patients who were re-irradiated with HDR-ISBT are summarized in Table 1.

**Table 1.** Recent studies on re-irradiation with HDR-ISBT for recurrent cervical cancer.

| Study | Cases with Previous RT (Total) | Primary Tumor Site (Case Number) | Treatment Regimen | Delivered Dose (Gy) | Local Control Outcomes | Other Outcomes | Toxicities | Prognostic Factors |
|---|---|---|---|---|---|---|---|---|
| Zolciak-Siwinska et al. (2014) [17] | 20 (20) | Cervix (19) + Vagina (1) | HDR BT alone (17) HDR BT + EBRT (2) HDR BT + EBRT + chemotherapy (1) | Re-irradiation EQD2: 48.8 Gy (16–91 Gy) Cumulative EQD2: 133.5 Gy (96.8–164.2 Gy) | 3-year LC 45% | 3-year OS 68% 3-year DFS 45% | Grade 3 late toxicity: n = 3 | Interval between radiations ≤ 12 months (LC, OS, DFS) Tumor diameter> 3 cm (LC, OS, DFS) |
| Mabuchi et al. (2014) [18] | 52 (52) | Cervix | HDR ISBT | Rx: 42 Gy/7 f | Response rate 76.9% | Median OS 32 m Estimated 5-year OS 52.6% | Grade 3–4 late toxicity: n = 13 | Tumor diameter ≥ 4 cm (OS) Primary tumor FIGO staging III–IV (OS) DFI ≤ 6 months (OS) |
| Mahantshetty et al. (2014) [19] | 30 (30) | Cervix | HDR ISBT alone (24) Intracavitary HDR BT alone (6) | EQD2: 42 Gy (37–46 Gy) | 2-year LC 44% | 2-year DFS 42% 2-year OS 52% | 2-year Grade 3 toxicity rate: 23% | Re-irradiation dose < 40 Gy EQD2 (LC) |
| Umezawa et al. (2018) [20] | 18 (18) | Cervix | HDR ISBT alone (13) EBRT + HDR ISBT (5) | EQD2: 62.5 Gy (48.6–82.5 Gy) | 2-year LC 51.8% | 2-year PFS 20% 2-year OS 60.8% | Grade 3–4 late toxicity: n = 3 | Hemoglobin level < 12.5 g/dL (LC) Tumor diameter ≥ 40 mm (LC) |
| Silva et al. (2019) [21] | 45 (45) | Cervix | HDR ISBT ± EBRT (4) ± chemotherapy (13) | Rx: 40–60 Gy/4–6 f | CR rate 67% | 1-year OS 71% 5-year OS 52% 5-year DFS (patients with CR) 45% | Grade 3–4 late toxicity: n = 15 | - |
| Raziee et al. (2020) [22] | 26 (26) | Endometrium (20) Cervix (4) Vulva (1) Vagina (1) | HDR ISBT | EQD2: 29.1 Gy (16.1–64.6 Gy) | 2-year LC 50% | 2-year PFS 38% 2-year OS 78% | Grade 3 late toxicity: n = 2 | - |
| Jiang et al. (2020) [23] | 27 (32) | Cervix (17) Endometrium (5) Vagina (6) Ovary (3) Vulva (1) | HDR ISBT | Rx: 10–36 Gy, 5–6 Gy/f, 2–6 f | 1-year LC 51.7% | Median TTF 15.4 months | Grade 3–4 late toxicity: n = 3 | - |

Abbreviations: LC, local control; OS, overall survival; PFS, progression-free survival; DFS, disease-free survival; CR, complete remission; Rx, prescription; EQD2, equivalent dose in 2 Gy per fraction; EBRT, external beam radiotherapy.

### 2.1.1. Efficacy

Silva et al. [21] conducted a retrospective cohort study on 45 patients with recurrent cervical cancer who underwent HDR-ISBT at a prescribed dose of 40–60 Gy/4–6 f, 2 f/d. The results showed that up to 30 cases (67%) achieved CR during a median follow-up time of 57 months. About half of the patients were alive 5 years after re-irradiation. The 5-year DFS rate of the 30 women with CR was 42%.

Mabuchi et al. [18] retrospectively analyzed 52 patients with recurrent cervical cancer after central pelvic radiotherapy who were treated with HDR-ISBT at prescribed doses of 42 Gy/7 f, 6 Gy/f, or 2 f/d. The response rate was 76.9%, in which 31 cases (60%) achieved CR. The median survival time was 32 months, and that was 47 months for patients achieving CR.

Umezawa et al. [20] reported 18 patients who were re-irradiated with HDR-ISBT after radical radiotherapy or postoperative radiotherapy for cervical cancer, combined with or without EBRT. The prescribed dose ranged from 2.5 to 6.0 Gy/f with a median CTV $D_{90}$ of 62.5 Gy $EQD_2$. The 2-year LC, PFS, and OS rates were 51.3%, 20%, and 60.8%, respectively.

Similar results had been reported by Raziee et al. [22]. They enrolled 26 patients with recurrent gynecological tumors who received interstitial BT as re-irradiation, including four cases of cervical cancer. The median prescribed dose of $D_{90}$ was 29.1 Gy $EQD_2$. Moreover, the 2-year LC and OS rates were 50% and 78%, respectively.

Jiang et al. [23] evaluated the feasibility and safety of HDR-IBT assisted with 3D-printed individual template (3D-PIT) for central pelvic recurrent gynecologic cancer. With a prescription dose of 10–36 Gy/2–6 f, the objective response rate (ORR) reached 84.4%. The median time-to-progression (TTP) was 15.4 months, and the 1-year LC rate was 51.7%.

Liu et al. [24] have recently suggested 3D-CT-guided HDR-IBT for patients with recurrent cervical cancer, in which needles were adjusted repeatedly using multiple CT scans until satisfactory dose distribution was achieved, and this technique may currently be clinically feasible. However, the long-term clinical outcomes remain to be further assessed.

In summary, HDR-ISBT for recurrent cervical cancer can achieve a CR rate of 60–76%, with 2-year LC and OS rates of over 40% and 50%, respectively. However, owing to the large heterogeneity and inconsistent treatment patterns, including HDR-ISRT alone, HDR-ISBT combined with ERBT, or chemotherapy, the outcomes of previous studies have not been comprehensively compared. However, compared with the remission rate of 20–50% [25–27] in previous studies using 2D-BT, a significant improvement was found in disease remission and LC after the emergence of 3D-image-guided BT. Moreover, the CT-guided MRI and positron emission tomography (PET)-guided BT should be utilized in clinical practice [28]. The efficacy and adverse effects of HDR-ISBT with different image-guided techniques may vary greatly. Further research is still warranted to clarify its efficacy and safety to regulate treatment procedures, dose prescription, and predictions of acute toxicity in OARs.

### 2.1.2. Prognostic Factors

Several studies have reported prognostic factors of the tumor patients treated by HDR-ISBT. Amsbaugh et al. [29] showed that tumor size was a significant predictor of poor PFS and OS (hazard ratio [HR, for 1 cm increase in tumor size], 1.61). Mabuchi et al. [18] demonstrated that the maximum tumor diameter ≥ 40 mm, primary tumor FIGO staging III–IV, and DFS ≤ 6 months were independently poor prognostic factors. Mahantsetty et al. [19] suggested that the LC rate was significantly higher in patients who were treated with a higher prescription dose (>40 Gy $EQD_2$ vs. ≤40 Gy EQD2, 52% vs. 34%). Umezawa et al. [20] pointed out that hemoglobin levels and the maximum tumor diameter are prognostic factors associated with the LC rate. In a study by Zolciak-Siwinska et al. [17], it was revealed that an interval of ≤12 months between irradiations and a tumor diameter > 3 cm negatively affected OS, DFS, and LC. Therefore, it is inferred that HDR-ISBT can be used in cases with a low tumor load and in the early stages of recurrence.

### 2.1.3. Toxicity

The incidence rates of adverse reactions reported by different centers remarkably vary. The common adverse reactions included vaginal ulcers, fistulae (rectovaginal and/or vesicovaginal), rectal bleeding, cystitis, pain, etc. A high incidence of adverse reactions was reported in the study by Silva et al. [21], which included 15 patients of grade 3–4 adverse reactions (33%) and 23 women (51%) of fistulae, while it was irrelevant to clinical remission. However, the incidence of fistulae is higher when chemotherapy is used, as suggested by the results of multivariate analysis. In a study by Mabuchi et al. [18], 25% of patients had grade 3–4 adverse reactions, including nine cases of vaginal fistulae. In a study by Mahantsetty et al. [19], the actuarial rate of grade 3 adverse reactions (including bowel, bladder, and vaginal fibrosis) was 23% over 2 years. Grade 3–4 adverse reactions were observed in three patients (16.7%) in a study by Umezawa et al. [20], which all occurred within 6–9 months after treatment. Late grade 3 adverse reactions were also observed in three patients (15%), as reported by Zolciak-Siwinska [17]. Grade 5 adverse reactions were reported in only one study, in which one patient developed post-implantation intestinal obstruction and died within 1 month after BT [30].

### 2.1.4. OAR Dose Constraints

The OAR dose constraints in re-irradiation depend on the previous dose, proximity, the size of both OARs and tumors, and interval between radiations. Abusaris et al. [31] demonstrated that for re-irradiation of the OARs, the maximum dose should be considered 50% more than the normal constraint. Therefore, Dmax of 100 Gy for rectum, 90 Gy for bowel, and 110 Gy for bladder are safe and can be used for re-irradiation. Taking the interval between two radiations and tissue repair into account, a dose reduction of 50% is permitted for re-irradiation 12 months after the last radiation, which is 25% for re-irradiation after 6–12 months and should be zero in case of re-irradiation within 6 months. This finding was also supported by Zolciak-Siwinska et al. [17], with a median interval (between two radiations) of 23 months. Of note, the dose was carefully calculated in reference for initial or re-irradiating treatment, particularly for 2D or 3D planning.

More specifically, for HDR-ISBT, according to a phase II trial of image-based HDR-ISBT by Martínez-Monge [30], when the prescribed dose was 38 Gy/8 bid fractions, the mean urethral dose should be lower than 115% of the prescribed dose (<5.5 Gy), and rectal $D_{10}$ and bladder $D_{19}$ should be lower than 70% (<3.3 Gy) and 80% (<3.8 Gy) of the prescribed doses, respectively. The dose reference given by Liu et al. [24] was HR-CTV $D_{90}$ (EQD2) $\geq$ 50 Gy, bladder $D_{2cc}$ (EQD2) $\leq$ 90 Gy, rectum $D_{2cc}$ (EQD2) $\leq$ 75 Gy, and sigmoid colon $D_{2cc}$ (EQD2) $\leq$ 75 Gy. Amsbaugh et al. [29] suggested that urethral $D_{0.1cc}$ was a predictive factor for grade $\geq$ 2 urethral toxicity (HR (for 1 Gy EQD2 increase), 1.156; 95% CI: 1.001, 1.335), while no urethral $D_{0.1cc}$ cutoff point was identified.

A systematic review by Bockel et al. [32] included 15 studies of image-guided BT for salvage therapy, and no association between OAR dose constraints and late toxicity was shown, which may be due to the small sample size. However, in studies with a high incidence of late toxicity $\geq$ grade 3, rectal and bladder $D_{2cc}$ reached 100 Gy.

Therefore, it is recommended that when the target dose is at least 40 Gy EQD2, two different methods of restricting doses for OARs can be considered, as described in the report of the American Association of Brachytherapy (ABS) Gynecological Cancer Reradiation Therapy Working Group [33]. The first method requires exceeding the classical cumulative dose of OAR (80–90 Gy for bladder EQD2 $D_{2cc}$ and 70–75 Gy for rectal and sigmoid EQD2 $D_{2cc}$), in order to guarantee HR-CTV $D_{90}$ of 60–65 Gy. The second method aims to maintain HR-CTV $D_{90}$ at >40 Gy, while strictly adhering to these OAR dose constraints. The potential risk of late toxicity is relatively high in the first strategy, in which the incidence of G3 toxicity is reported to be $\geq$15% by most studies using the first method.

### 2.2. Permanent Radioactive Seed Implantation (PRSI)

PRSI, a low-dose-rate BT (LDR-BT), aims to place a microscopic radioactive source directly into or around the tumor to kill tumor cells by continuous emission of radiation from radionuclides. Compared with HDR-ISBT, only one single interstitial insertion is required, in which the radioactive source is permanently retained in the body. PRSI has the dosimetric advantages of high local dose and minimal overlap into adjacent normal tissues, and it can be used in in-field re-irradiation of well-selected patients. In PRSI, nucleoids such as $^{125}$I, $^{103}$Pd, $^{131}$Cs, and $^{198}$Au are extensively utilized. During the past century, an important development in permanent implants was the introduction of interstitial seeds with moderately long half-lives of 10–60 days that emit a cascade of low energy (20–40 keV) characteristic X-rays and γ-rays [34]. Compared to $^{198}$Au (half-life: 2.7 days, mean energy: 412 keV), $^{125}$I (half-life: 59.4 days, mean energy: 28.37 keV), $^{103}$Pd (half-life: 16.99 days, mean energy: 20.74 keV), and $^{131}$Cs (half-life: 9.7 days, mean energy: 30.4 keV) are characteristic with longer half-life, lower energy, and higher radiation safety [35,36]. Moreover, it has been suggested that "dual-isotope", a mixture of different isotopes, such as $^{125}$I combining with $^{103}$Pd, might contribute to a better radiotherapy outcome by optimizing the dose distribution [37–39]. However, specific protocols need to be established concerning putting "dual-isotope" into clinical use widely.

At the present, an optimum radionuclide for treating recurrent cervical cancer has not been defined. Re-irradiation by PRSI is typically performed with $^{125}$I. Radioactive $^{125}$I seed implantation (RISI) is a standard treatment modality for early-stage low-risk prostate cancer and has also been used in the treatment of various recurrent solid tumors, owing to its optimal dose profile, low invasiveness, and repeatability [40–43]. Early studies have used this modality for the treatment of recurrent cervical cancer. Sharma et al. [44] demonstrated the efficacy and safety of interstitial implantation of $^{125}$I seeds in 40 cases of recurrent gynecologic malignancies after the previous radiotherapy. The tumor control rate reached 67%, and 33% of patients had a DFS of more than 2 years.

Traditionally, seed implantation was performed without imaging guidance, and its efficacy highly depended on an operator's experience and technique, plus preoperative planning, and postoperative verification. Although the tumor control rate is acceptable, the long-term efficacy is poor, and the efficacy reported in different studies varies widely due to the lack of established prescription dose regulation. Due to the development of imaging technology in BT, image-guided radioactive $^{125}$I seed (IGRIS) implantation allows for more precise treatment of recurrent cervical cancer and dynamic observation of the puncture path pre- and intraoperatively to ensure high conformality. With the image guidance, a 3D digital individualized body-shaped template is created according to preoperative CT scan. After the preoperative planning is settled, a real-time intraoperative CT scan was performed to monitor the needle's position during RISI, and the postoperative scan was conducted after seed implantation immediately to verify dose distribution of seeds (Figure 1a–d).

Recently, a Chinese consensus was proposed by Jiang et al. [45]. Nevertheless, there is currently no prospective study on the prescribed dose of permanent seed implantation, except for prostate cancer. Therefore, based on the treatment guidelines of prostate cancer, relevant clinical evidence, and clinical experience, the current treatment dose is recommended as follows: seed activity of 0.4~0.5 mCi; GTV $D_{90}$ of 110~130 Gy, which can be externalized by 5~6 mm to form a CTV $D_{90}$ of 90~110 Gy; and OARs dose constraints: intestinal $D_{2cc}$ < 100% of the prescribed dose, $D_{0.1cc}$ < 200 Gy, urethral $D_{10}$ < 150%, and $D_{30}$ < 130% of the prescribed dose.

CT-guided $^{125}$I seed implantation was performed by Tong et al. [46] for the treatment of 35 recurrent pelvic lesions (33 patients), with a prescribed dose of 90–150 Gy. During a median follow-up of 16 months, the median local tumor progression-free survival (LTPFS) reached 7 months, and the median OS time reached 12 months. Qu et al. [47] retrospectively analyzed 36 patients with pelvic recurrent cervical cancer (PRCC) who were treated with IGRIS. During a median follow-up of 11.5 months, the tumor control rate was 89%. Moreover, the 1-year LPFS rate and OS rate reached 34.9% and 52.0%, respectively. Liu et al. [48]

reviewed a series of 103 patients who recurred with inoperable cervical cancer after prior radiotherapy. A total of 111 lesions underwent RISI, 75 of which were pelvic peripheral recurrences, with a median prescription dose of 120 Gy. The 3-year LC and OS rates were 75.1% and 20.8%, respectively, with a median OS of 17 months. Late toxicity was observed in two patients with rectovaginal fistula. Recent studies on re-irradiation with PRSI for recurrent cervical cancer are listed in Table 2.

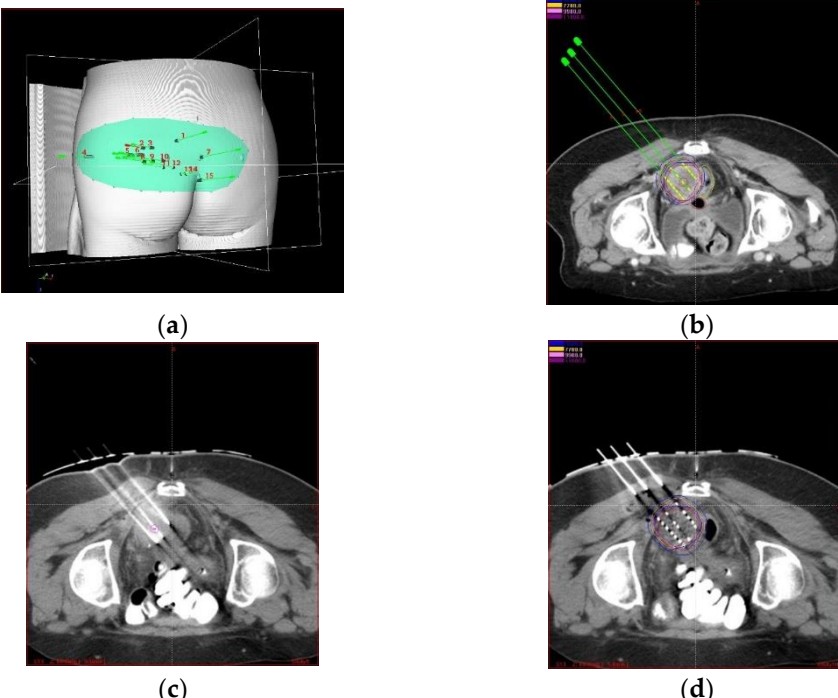

**Figure 1.** (**a**) 3D digital body-shaped template according to preoperative CT image. (**b**) pre-operative planning based on CT scan. (**c**) intraoperative and (**d**) postoperative validation based on CT scan.

In general, the advantages of CT-guided $^{125}$I seed implantation include high resolution, real-time image reconstruction, and contrast scanning, especially for re-irradiation with a higher demand of more precise treatment modalities. It is suggested that this technique is more appropriate for peripheral recurrence than central recurrence. On the one hand, seeds could easily fall off from the vagina in patients with central recurrence, which leads it hard to remain in the thin vaginal wall. On the other hand, the central lesions are easily spread along the vaginal wall, leading to difficulty in the confirmation of tumor boundary. Moreover, pelvic sidewall recurrences are located relatively far from OARs. Therefore, a high dose could easily be delivered to the target site, meeting the OAR dose constraints. Meanwhile, there are still a number of challenges. Firstly, it relies heavily on an operator's experience, and despite the image guidance, it is difficult for the insertion to be consistent with the pre-plan. Moreover, for the maximum consistency with the preoperative plan, the average operation time is about 2–3 h, with the patients excessively exposed to radiation.

**Table 2.** Recent studies on re-irradiation with PRSI for recurrent cervical cancer.

| Study | Cases with Previous RT (Total) | Primary Tumor Site | Recurrent Tumor Site | Interval between Radiations (Month) | Treatment Regimen | Re-Irradiation Dose (Gy) | Tumor Size | Median Seeds Number | Median Follow-Up Time | Local Control Outcomes | Other Outcomes | Toxicities | Prognostic Factors |
|---|---|---|---|---|---|---|---|---|---|---|---|---|---|
| **Lei et al. (2017) [49]** | 17 (17) | Cervix | Cervix Vaginal cuff Pelvic LNs Extra-pelvis | n/s | CT-guided $^{125}$I seed implantation $\pm$ chemotherapy | Matched peripheral dose: 145 Gy | $0.5 \times 0.5$ cm–$5 \times 6$ cm | 20 (6–68) | 9.5 (4–18) | Overall response rate 58% | 1-year OS 18.3% | Grade 3–4 late toxicity: n = 0 | n/s |
| **Tong et al. (2017) [46]** | 33 (33) | Cervix | Cervix | n/s | CT-guided $^{125}$I seed implantation + chemotherapy | Rx: 90–150 Gy | n/s | 50 (20–95) | 16 | 1-year LC 55.5% | 1-year OS 65.5% 2-year OS 43.6% | Grade 3–4 late toxicity: n = 2 | Tumor diameter < 4 cm (LC) D90 $\geq$ 130 Gy (LC) Good performance status (OS) |
| **Qu et al. (2019) [47]** | 36 (36) | Cervix | Pelvic Sidewall (21) Central Pelvis (15) | 12 (2–60) | CT-guided $^{125}$I seed implantation | GTV D$_{90}$: 128.5 $\pm$ 47.4 Gy | 59.2 cm$^3$ (2.5–116.5 cm$^3$) | 62.5 (10–140) | 11.5 (2–30) | 1-year LPFS 34.9% 2-year LPFS 20% | 1-year OS 52% 2-year OS 19.6% | Grade 3–4 late toxicity: n = 1 | Pathological type (OS) recurrence site (OS, LPFS) lesion volume (LPFS) D90 $\geq$ 105 Gy (LPFS) |
| **Liu et al. (2019) [48]** | 103 (103) | Cervix | Pelvic Sidewall (75) Central Pelvis (8) Extra Pelvis | 11 (2–70) | 3D-PNCT assisted CT-guided $^{125}$I seed implantation | Rx: 120 Gy (100–180 Gy) | GTV 37.7 cm$^3$ (2.6–237.8 cm$^3$) | 63 (8–186) | 12 (2–43) | 1-year LC 87.4% 3-year LC 75.1% | 1-year OS 68.1% 3-year OS 20.8% | Grade 3–4 late toxicity: n = 2 | Pathological type (LC, OS) Hemoglobin level (LC, OS) D90 $\geq$ 130 Gy (LC, OS) Recurrence site (OS) |
| **Chen et al. (2020) [50]** | 23 (32) | Cervix (11) Non-Cervix (21) | Retroperitoneal lymph nodes | n/s | 3D-PNCT assisted CT-guided $^{125}$I seed implantation | Rx: 140 Gy (115–160 Gy) | n/s | 62.5 (15–197) | 15.3 (9.2–33.5) | 1-year LC 66.2% 2-year LC 43.2% | 1-year OS 74.1% | Grade 3–4 late toxicity: n = 0 | Univariate analysis: Tumor size $\leq$ 49.8 cm$^3$, D$_{90}$ < 130 Gy or D$_{100}$ < 63 Gy (LC) |

Abbreviations: LC, local control; OS, overall survival; LPFS, local progression free survival; Rx, prescription; CT, computed tomography; 3D-PNCT, three dimensional-printed noncoplanar template; D90, dose delivered to 90% of the target volume; D100, dose delivered to 100% of the target volume; n/s, not specific.

## 3. SBRT

SBRT is defined as a tissue-sparing ultra-hypofractionated RT (usually ≤5 fractions) along with utilizing imaging precision (e.g., 4D-CT, SGRT, RPM etc.) which could be performed by standard linear accelerator or dedicated machines such as CyberKnife. SBRT is characterized by a high level of conformality, high doses of radiation, and a relatively short treatment course. Previous studies have reported SBRT in gynecologic neoplasms, mainly for inoperable small pelvic wall lesions and isolated pelvic or para-aortic lymph node recurrences [51–54]. Recent studies have reported the outcomes of recurrent cervical cancer treated by SBRT, which are shown in Table 3.

Seo et al. [55] performed SBRT on 23 patients with recurrent cervical cancer at the pelvic sidewall, of whom 17 cases received re-radiotherapy, with a median dose of 39 Gy/3 f. The 2-year survival rate, local progression-free survival (LPFS) rate, and progression-free survival (PFS) rate were 43%, 65%, and 52%, respectively. Additionally, three patients developed rectovaginal fistula. It was reported that gross tumor volume (GTV) < 30 cc was an independently favorable prognostic factor, which was also confirmed by the findings of Hasan et al. [56] that a smaller clinical tumor volume (CTV) were favorable prognostic factors. In a study of Park et al. [57], 85 patients (100 lesions) with recurrent or oligometastatic cervical cancer were treated by SBRT with a median dose of 39 Gy/3 f and a biologically effective dose (BED) of 90 Gy. Among them, 71 patients were treated with re-irradiation, with a median BED of 79 Gy. The 2- and 5-year local control (LC) rates were 85% and 79%, respectively. The 2- and 5-year survival rates were 57.5% and 32.9%, respectively. Additionally, five cases had grade 3–4 late adverse reactions. In another study [58], SBRT was used for re-irradiation of five patients with locally recurrent cervical cancer at a prescription dose of 15–20 Gy/3~4 f. The results showed that three and two patients achieved complete response (CR) and partial response (PR), respectively.

To date, cases with non-cervical gynecological cancer, oligo-metastasis, and persistent diseases have been mainly reviewed, and it is thus difficult to indicate the proportion of SBRT in the treatment of recurrent cervical cancer. Hasan et al. [56] reported their experience in the treatment of 30 pelvic or isolated para-aortic recurrent lymph nodes of 11 cervical cancer patients via SBRT. The results showed that the five-year survival rate was 42%, with a median survival of 43.4 months. In addition, 20 patients with isolated pelvic or intra-abdominal recurrence of gynecologic malignancy within a previously irradiated field were re-irradiated by Ling et al. [59]. The 2-year OS, LPFS, and DPFS rates were 43%, 65%, and 52%, respectively, and the incidence of ≥grade 3 late toxicities was 14.3%. According to a systematic review [60], a total of 73 patients with pelvic recurrence of gynecologic tumors, who were treated by SBRT with a median BED of 22.5 (range, 12–61.7) Gy, were involved in 10 studies. The LC rate was 86%, and 19.2% of patients experienced grade 3 to 4 complications.

In a survey recruiting 11 experienced oncologic radiologists [61], the majority of experts considered a salvage radiotherapy (reirradiation or non-reirradiation) for lymph node recurrence (81%) and recurrent gynecologic tumors at the primary site (91%) when BT was not feasible in either definitive or boost treatment. Most participants in this survey pointed out that during the treatment of nodal recurrence, CTV should be considered as a gross nodal lesion with an expansion, and GTV only or GTV with an expansion was recommended for recurrence at the primary site. Meanwhile, the reference dose/fraction regimen for definitive SBRT is as follows: (a) for lymph node recurrence: Equivalent Dose In 2-Gy Fractions (EQD$_2$) = 36 Gy (15.6–60 Gy)/3–5 f; (b) for recurrence at the primary site: EQD$_2$ = 40.4 Gy (27–71.2 Gy)/3–5 f. For boosting SBRT after standard radiation, a median dose of 36.75 (range, 15.6–60) Gy is recommended. Dose constraints in this survey were variable, while dose-constrains were mainly considered for small intestine, which could be irradiated with Dmax < 35–39 Gy or V25 Gy < 5 cc in 3-fractions treatment and D$_{max}$ < 15–25 Gy or D$_{2cc}$ < 20 Gy in 5-fractions treatment.

**Table 3.** Recent studies on re-irradiation with stereotactic body radiation therapy (SBRT) for recurrent cervical cancer.

| Study | Nature of the Study | Cases with Previous Radiotherapy (Total) | Primary Tumor Site | Recurrent Tumor Site | Treatment Regimen | Median Re-Irradiation Dose (Gy) | Median GTV (cm³) | Local Control Outcomes | Other Outcomes | Toxicities | Prognostic Factors |
|---|---|---|---|---|---|---|---|---|---|---|---|
| **Park et al. (2015)** [57] | Retrospective | 71 (85) | Cervix | Abdominopelvic lymph nodes | SBRT | Rx: 39 Gy/3 f (BED, 89.7 Gy) (44 cases) BED, 79.2 Gy (re-irradiation group) | n/s | 2-year LPFS 82.5% 5-year LPFS 78.8% | 2-year OS 57.5% 5-year OS 32.9% | Grade 3–4 late toxicity: n = 5 | BED ≥ 89.7 Gy and 69.3 Gy (LC) para-aortic LN vs. pelvic LN (LC) disease-free interval ≥ 36 months (LC, OS) |
| **Seo et al. (2016)** [55] | Retrospective | 17 (23) | Cervix | Pelvic sidewall | EBRT + SBRT boost + chemotherapy (7) SBRT + chemotherapy (14) SBRT (2) | Rx: 39 Gy (27–45 Gy)/3 f | 40 (2–215) | 2-year LPFS 65% | 2-year OS 43% 3-year OS 27% 2-year DPFS 52% | Grade 3–4 late toxicity: n = 3 | GTV < 50 cm³ (LC) GTV < 30 cm³ (OS) |
| **Pontoriero et al. (2016)** [58] | Retrospective | 5 (5) | Cervix | Central pelvis | SBRT | Rx: 15–20 Gy/3–4 f | 20 (8.2–47.4) | n/s | CR: n = 1 PR: n = 2 PD: n = 1 OC: n = 1 | Grade 3–4 late toxicity: n = 0 | n/s |
| **Ling et al. (2019)** [59] | Retrospective | 20 (20) | Cervix (6) Uterus (11) Vagina (1) Ovary (1) Vulva (1) | Pelvis (13) Para-aortic nodes (6) Celiac nodes (1) | SBRT ± chemotherapy (17) EBRT + SBRT boost ± chemotherapy (3) | Rx: 44.5 Gy (33.8–45 Gy) $BED_{10}$, 82.7 Gy (64.1–85.5 Gy) | 9.7 (4.6–35.9) | 3-year LC 61.4% | 3-year DPFS 44.0% 3-year OS 51.9% | Grade 3–4 late toxicity: n = 3 | n/s |

Abbreviations: LN, lymph node; GTV, gross tumor volume; LC, local control; OS, overall survival; LPFS, local progression-free survival; DPFS, disease progression-free survival; CR, complete response; PD, partial response; PD, progression disease; OC, other cause of death. Rx, prescription; BED, biological equivalent dose; EBRT, external beam radiotherapy; n/s, not specific.

To date, SBRT cannot be considered as a substitute for BT and is mainly used in patients not indicative for surgery and BT. In most settings, SBRT is regarded as a curative-intended first-phase salvage therapy or as a boost therapy after EBRT when lesions are not amenable to BT. Furthermore, it is better tolerated when it is delivered to the nodal recurrence site, while the incidence of toxicities may increase in the SBRT treatment of non-nodal recurrence [62]. Mendez et al. [60] found that SBRT was associated with a high incidence of gastrointestinal toxicity (approximately 20%) in patients with non-nodal pelvic recurrence. However, the incidence of toxicities was similar or higher than that in patients who received interstitial BT (ISBT) or pelvic exenteration as a salvage therapy [18,63]. Notably, the SBRT studies discussed above are all retrospective institutional series with small number of cases and short period of follow-up, lacking meaningful clinical end points and supportive outcomes. Hence, SBRT is sometimes attempted in a small quantity of centers, and a strict control of indications and toxicities is still required by radiotherapists.

## 4. Prospects

### 4.1. RISI-Assisted 3D Printed Template (RISI-3DPT)

As mentioned above, $^{125}$I seed permanent implantation for recurrent cervical cancer has achieved satisfactory results. However, this conventional technique may be disadvantaged by a biased judgment of the insertion angle, long operation time, radiation risk caused by multiple CT scans, and uneven dose distribution due to the irregular shape of the stump. Given these deficiencies, 3D-printed minimally invasive guidance templates have been considered for $^{125}$I seed implantation in recent years. This technique uses 3D digitization to create a personalized template, containing puncture path information for the designed protocol, which can be used to preoperatively plan to ensure compliance and accuracy of the target. The 3D-printed templates are divided into 3D-printed coplanar templates (3DPCTs) and 3D-printed non-coplanar templates (3DPNCTs) (Figure 2a–d), and studies have demonstrated the utilization of 3D-printed templates in RISI [64–67]. A meta-analysis [68] showed that 3D-printed template-assisted $^{125}$I seed implantation optimizes dose distribution, while reducing operation time, compared with conventional free-hand implantation. Compared with 3DPCTs, 3DPNCTs can increase the high dose volume within the target, including the increased $V_{150}$ and $V_{200}$, with fewer puncture stitches and higher safety in puncture routes.

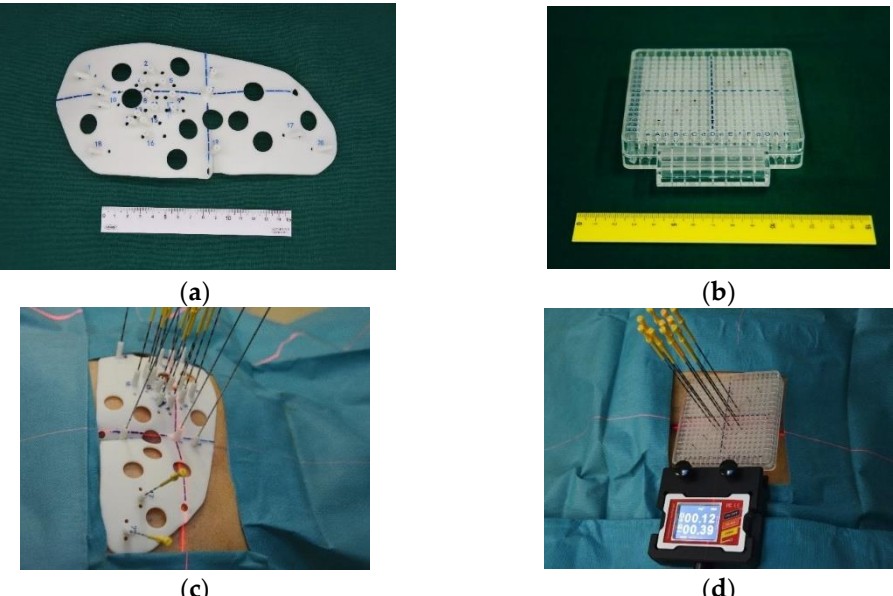

(a)　　　　　　　　　　　　　(b)

(c)　　　　　　　　　　　　　(d)

**Figure 2.** (**a**) Real product of 3DPNCT and (**b**) 3DPCT. (**c**) 3DPNCT and (**d**) 3DPCT immobilized with stable needles in seed implantation for recurrent cervical cancer.

A recent study concentrated on the variability between pre- and postoperative planning dosimetric parameters for the application of 3D-PNCT assisted CT-guided [125]I seed implantation therapy in patients with non-central recurrence of gynecologic cancer [69]. The results showed that the mean inserted depth deviation of all 499 needles was $0.8 \pm 1.0$ cm. The mean angular deviation was $2.2 \pm 2.1°$ with a mean needle tip distance deviation of $0.4 \pm 0.3$ cm. There was no significant difference between preoperative and postoperative $D_{90}$, $D_{100}$, $V_{100}$, $V_{150}$, $V_{200}$, and HI.

It is suggested that 3D-printed minimally invasive-guided template-assisted treatment for recurrent cervical cancer has significant dosimetric advantages over conventional free implantation, with a short implantation time and a reliable repeatability. Moreover, 3D-PNCT is able to fit the body surface contour perfectly. The position, angle, and direction of a needle can be reversely designed to meet the target dose requirements and avoid damage to OARs. Meanwhile, 3D-PNCT obtains a coordinate as a reference for fixation with the body, resulting in improved precision and accuracy. Additionally, 3D-printed template-assisted [125]I seed implantation for re-irradiation of cervical cancer has also initially shown a better efficacy, either for intra-pelvic or extra-pelvic lesions, as was shown in the studies by Liu et al. [48] and Chen et al. [50], in which the 1-year LC rate was 66.2–87.4% and the 1-year OS rate reached within 70%.

In conclusion, the application of 3D-PNCT reduces the dependence of RISI on an operator's experience, shortens treatment time, attenuates operational risks, and makes RISI easier to learn and promote.

### 4.2. Imaging Navigation System-Assisted RISI

As PRSI heavily relies on an operator's experience, it is plausible that the quality control is likely to vary largely. The introduction of an image navigation system provides better accuracy and feasibility, which tracks the needle position in real-time by a tracer to locate the lesion on the fusion image, guiding the puncture procedure. In numerous studies, it has been used for the treatment of solid tumors using seed implantation, with a great puncture accuracy [70,71]. Moreover, Ji et al. [72] suggested that there was no significant difference in dosimetric parameters between post-operative and preoperative plans, regardless of the type of organ, including head and neck, chest wall, and abdominopelvic lesions. In summary, the integration of multiple advanced technologies, such as an artificial intelligence navigation system, will additionally promote the popularization of PRSI, and several clinical trials are underway.

### 5. Conclusions

At the present, pelvic exenteration $\pm$ intraoperative radiotherapy is preferred for patients with central pelvic recurrence after radiotherapy. Such patients may be treated with secondary radiotherapy and/or chemotherapy. Due to the advantages of non-invasiveness, high precision, and simplicity, SBRT has been used for re-irradiation of patients with laterally recurrent cervical cancer and very small lesions in some studies, and a high LC rate and a low incidence of severe toxicity are achieved. However, to date, the relevant studies are still limited, which are all retrospective studies with small sample sizes. BT has also been utilized to treat previously radiated PRCC, mainly including HDR-ISBT and PRSI, and has shown promising efficacy and safety; HDR-ISBT is more appropriate for patients with central pelvic recurrence, while RISI is highly recommended for patients with peripheral pelvic recurrence. With the assistance of an imaging navigation system and 3D-PT, the efficacy, safety, and accuracy of PRSI are further improved, and the difficulty of learning and promoting PRSI is reduced. However, expertise and advanced hospital equipment are still needed to ensure [125]I seeds are inserted accurately, and the training period for professionals is long. Therefore, the indications, dose regulations, and OAR dose constraints for re-irradiation of recurrent cervical cancer remain to be further clarified.

**Author Contributions:** Conceptualization: J.W.; Project administration: J.W.; Resources: Z.S., P.J., A.Q., Y.J. and H.S.; Writing—original draft: Z.S. and A.Q.; Writing—review & editing: P.J., A.Q. and J.W. All authors have read and agreed to the published version of the manuscript.

**Funding:** This research received no external funding.

**Institutional Review Board Statement:** Not applicable.

**Informed Consent Statement:** Not applicable.

**Conflicts of Interest:** The authors declare no conflict of interest.

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
