# Peer review of "Re-Irradiation for Recurrent Cervical Cancer: A State-of-the-Art Review"

_curroncol, doi:10.3390/curroncol29080418_

Round 1

Reviewer 1 Report

Re-irradiation for Recurrent Cervical Cancer: A State-of-the-art

General comments:

The authors of the manuscript presented a comprehensive review within the context of re-irradiation for recurrent cervical cancer, aiming to report some core radiotherapy strategies that might be considered for managing recurrent cervical cancer such as SABR and brachytherapy. Followed by a detailed literature review presentation of some important aspects that related to the theme of the manuscript such prognostic factor, toxicity and OAR radiation dose tolerances.

The manuscript is well-written, with an excellent introductory and discussion sections. The subject discussed is highly topical and covers an essential subject for the radiation oncology community. Nonetheless, the manuscript suffers from minor flaws that the authors must address prior to the publication.

Specific comments:

1)             Chronology of treatment options, the manuscript has to be re-structured in such a way that brachytherapy section has to be presented prior to the SABR section. Please amend the manuscript accordingly.

2)             Line 64-65, the authors has stated that “SBRT is defined as an external radiation therapy technique delivering high doses of radiation to the target tumour in a higher-dose fractionated modality with high precision”.

·         Not really sure if you would define SBRT this way, seems moderate hypofractionation (e.g., 2.5-3.0 Gy) is considered SBRT by applying this definition.

·         Should be Ultra Hypofractionated RT along with utilizing imaging precision (e.g., 4D-CT, SGRT, RPM etc…).

3)             A) Table 1: The displayed SBRT studies are retrospective institutional series with very small number of patients and short period of follow up. Lack of meaningful clinical end points. Many readers don’t go through the references in greet details or don’t have access to them, and it is crucial to avoid sending a wrong message. Please discuss the limitations of the cited papers in table 1.

·         Table (1) … Please add a column to describe the nature of the paper (e.g., retrospective, prospective etc…)

·         Interesting median dose of 39 Gy/3f. usually RCT phase II (David Palma) is the best available evidence with doses ranging from 30-40 Gy in 5 fractions.

4)             Line 67-68, the authors have stated that “Previous studies have reported SBRT in gynecologic neo-67 plasms, mainly for inoperable pelvic wall lesions, larger central lesions that could not be 68 treated by brachytherapy (BT), and isolated pelvic or para-aortic lymph node recur-69 rences”

·      I would like to point out that pelvic tumours cannot be treated by brachytherapy if your radiotherapy centre has the capability to do hybrid and interstitial brachytherapy. Might be true for only simple intracavitary brachytherapy.

·      In fact, larger pelvic only recurrences are served better with full interstitial implant with better local control and less toxicity. Can you please clarify.

5)             Line 35-37, the authors have stated that “the central type refers to the recurrent lesion located at the centre or midline of the pelvis, which could invade anteriorly, posteriorly (bladder, rec-36 tum) or laterally (vaginal vault), while it does not reach the pelvic wall”. 

·         Using such a definition would exclude the most important anatomical landmark in the cervix, the parametria being considered central or can be treated with ISBT.  Additionally, vaginal vault is a superior structure (top of Vagina post-surgery). Lateral Cervix is always treated with no surgery and less likely to use vaginal vault language Please amend the manuscript accordingly.

6)             Line 134-135, “HDR-ISBT is a type of BT, where a needle is inserted directly into the tumour”. 

·         Reconsider the definition of ISBT: Not necessarily inserting catheters into tumours only. Some definitions such as inserting needles or catheters into the volume of interest can be utilized.

7)             Line 139-141, “Although HDR-ISBT is more advantageous than intracavitary irradiation for the treatment of bulky, deeply in-140 volved tumors, it is still clinically restricted for central recurrences”.

·         Please be careful with such a statement- Not restricted to central recurrence especially with what was defined as central in the introduction- meaning parametria cannot be treated!

·         Additionally, bulky word only used with de novo cervix cancer but not recurrence. Please clarify.

8)             In section 3.2, line 265-567, the LDR radioactive sources have been presented briefly. Additional physical properties of the seeds must be discussed.

For instance, Cs-131 has an average energy of 30.4KeV and half-life of 9.7 days and is considered to combine the advantages of the higher energy of I-125 and the high dose rate of the Pd-103. Some radiobiological advantages for each isotope over other have been claimed in literature. It has been recognised that there might be a role for a mixture of I-125, Pd-103 and Cs-103, which may provide a better radiobiological outcome. This technique is well known as dual-isotope.

9)      Other toxicities such sacrum necrosis and rectum perforation have to be discussed in section 3.3.1 Toxicity.

Author Response

Thanks for taking your time to review our manuscript. Please see the attachment.

Reviewer 2 Report

The authors wrote a review about different therapeutic options for recurrent cervical cancer. Interesting review on recurrent cervical cancer; all major therapeutical approaches are included such as sbrt, interstitial HDR brachitherapy and LDR permanent implants.

The review is well written but due to the nature of the research their results seem inconclusive.

I think the authors should try to perform a statistical analysis on the data they report in order to compare results from sbrt and brachytherapy.

References are appropriate .

Author Response

Thanks for taking your time to review the manuscript. Please see the attachment.

Reviewer 3 Report

The referred manuscript is a detailed revision of an interesting topic. In my  opinion there is an imbalanced proportion between the different strategies and approaches. For example, IORT review is too short and the opposite for PRSI. I understand that the authors have experience in sedd implant but the technique is by now not widely prescribed everyvhere.

Author Response

(The authors gave the same response as above.)
